# The pupil responds spontaneously to perceived numerosity

Elisa Castaldi[1,2], Antonella Pomè[2], Guido Marco Cicchini[3], David Burr [2,4✉] & Paola Binda [1]

Although luminance is the main determinant of pupil size, the amplitude of the pupillary light response is also modulated by stimulus appearance and attention. Here we ask whether perceived numerosity modulates the pupillary light response. Participants passively observed arrays of black or white dots of matched physical luminance but different physical or illusory numerosity. In half the patterns, pairs of dots were connected by lines to create dumbbell-like shapes, inducing an illusory underestimation of perceived numerosity; in the other half, connectors were either displaced or removed. Constriction to white arrays and dilation to black were stronger for patterns with higher perceived numerosity, either physical or illusory, with the strength of the pupillary light response scaling with the perceived numerosity of the arrays. Our results show that even without an explicit task, numerosity modulates a simple automatic reflex, suggesting that numerosity is a spontaneously encoded visual feature.

[1] Department of Translational Research and New technologies in Medicine and Surgery, University of Pisa, Pisa, Italy. [2] Department of Neuroscience, Psychology, Pharmacology and Child Health, University of Florence, Florence, Italy. [3] Institute of Neuroscience, National Research Council, Pisa, Italy. [4] School of Psychology, University of Sydney, Camperdown, NSW, Australia. ✉email: davidcharles.burr@unifi.it

Being able to efficiently estimate the number of enemies or prey is essential for survival: all animals[1]—from insects to humans—are capable of some form of numerosity discrimination. Much evidence suggests that numerosity discrimination is a basic and spontaneous sense, often referred to as the number sense[2,3]. For example, in monkeys and crows, single neurons are tuned to numerosity, before any training on numerosity tasks[4–6]; in humans, crude number discrimination has been documented as early as a few hours after birth[7,8]. Neural representation of numerosity is organized within a topographic principle, such as that for most sensory attributes[9,10]. Visual processing of numerosity is rapid, with numerosity-specific signals emerging in the occipital cortex only 75 ms after stimulus onset[11], and humans can saccade rapidly to the more numerous target, as quickly as 190 ms[12], implicating primitive, possibly subcortical, circuitry that quickly transforms numerosity information into an oculomotor response. This and other literature point to numerosity being a salient perceptual feature, eliciting a spontaneous perceptual response.

The pupillary light response is one of the most basic sensory responses, serving primarily to regulate light entry and aid dark adaptation[13,14]. However, even when luminance is kept constant, pupil size can vary predictably and systematically with the effective strength of stimulation. For example, attending to a bright or dark patch will enhance the pupillary response evoked by the patch[15,16]. Moreover, brightness[17] and size[18] illusions, and even the implied brightness of images of the sun and moon[19,20], can elicit strong and reliable pupillary constriction. That pupil size is susceptible to attention, visual illusions, and to the semantic content of images suggests that the subcortical structures that control pupil size receive modulatory signals from higher-level areas[21,22].

Here we exploit the pupillary light response to study the spontaneous nature of numerosity perception and show that it scales with perceived numerosity. Pupils constrict more to passively observed white arrays and dilate more to black arrays for more numerous (luminance matched) stimuli, whether perceived numerosity was manipulated by dot number or by exploiting a grouping-based illusion. This suggests an implicit association between numerosity and perceptual strength, which can be read out—objectively and quantitatively—from the pupil.

## Results

**The connectedness effect**. We measured pupil size, while participants passively viewed arrays of dots of different numerosities, real or perceived, leveraging on a strong numerosity illusion. Figure 1A, B show examples of the white versions of the stimuli: the experiments used both dark and bright stimuli, black and white dots and lines on a gray background. Using two versions of the connectedness illusion (panels A and B) helps to dismiss various potential artifacts (discussed later).

The stimuli varied both in physical and perceived numerosity: they comprised either 18 or 24 dots, and in half of the stimuli perceived numerosity was reduced by connecting pairs of dots to form dumbbells, a well-known illusion[23–25], obvious on inspection of Fig. 1. In the other half, the connecting lines were either displaced to random positions between the dots (experiment 1) or removed entirely (experiment 2). In both experiments, all stimuli had the same total number of white or black pixels, irrespective of numerosity and connectedness, covering 92.7 deg² for experiment 1 and 82.3 deg² for experiment 2. The area, defined as convex hull, was 513 deg² for all stimuli. Figure 1C, D show the Fourier transforms of the stimuli, discussed later.

After completing the first pupillometry experiment (where they were completely naive of the goals of the experiment), participants

were asked to judge which of the two sequentially presented arrays appeared to comprise more dots. One was the standard (18 isolated dots), the other the probe (connected or isolated: see "Methods" for experimental details). Figure 2 shows example psychophysical functions for the two experimental conditions (displaced or removed dots) in a typical participant. The median of the functions (0.5 response) yields the point of subjective equality (PSE), the numerosity of the probe that matched the reference. For the displaced-lines condition, the PSE was around 13, 28% less than for measurements with the isolated dots. For the removed-lines condition, the PSE changed a little less, to 14, about 23%. Figure 2B, D show the results averaged over all participants. The average bias in the connected patterns was about 30% for displaced-line stimuli and 22% for removed-line stimuli, similar to that reported by previous studies[23–25]. Given the magnitude of the effect, we would expect the perceived numerosity of a 24 connected-dot pattern to be around 17 in experiment 1 and 19 in experiment 2, similar to that of the unconnected 18-dot pattern.

**Pupillary light and dark responses**. We recorded pupil size while participants passively observed the stimuli of Fig. 1, which were repeatedly displayed for 6 s. Trials with isolated and connected stimuli were intermingled in pseudo-random order within the same session, with separate sessions for different colors and numerosities. We ran two separate experiments, using the two types of isolated controls, with connecting lines displaced to random positions, or removed.

Figure 3A shows the average baseline-corrected pupillary responses for experiment 1 (displaced-lines stimuli), to all stimulus types, both white and black. As expected, the pupil constricted for white-dot stimuli and dilated for black-dot stimuli. The light-evoked constriction was predictably faster than the dark-evoked dilation[21], but combining the two (by subtracting the light from the dark response) yielded a strong and sustained luminance response over the 6 s stimulus presentation (Fig. 3B). Importantly, although the total number of pixels (hence luminance) was always the same in all four conditions, the amplitude of the pupil-size modulation clearly varied across conditions.

We used the mean pupil difference (Fig. 3B) over the 1–6 s interval to index the pupil response strength. These values are shown in Fig. 3C (for individual data, see Supplementary Fig. 1). The stimulus with the highest perceived numerosity (24 isolated dots) elicited the strongest response, and that with the lowest perceived numerosity (18 connected dots) elicited the weakest response (paired t-test comparing 24 isolated vs. 18 connected: $t(15) = 4.5$, $p < 0.001$, Cohen's $d = 1.1$, $\log_{10}BF = 1.9$); 18 isolated and 24 connected dots, which had similar apparent numerosity, elicited an intermediate response. Thus, the strength of the pupillary luminance response depended on numerosity, both the physical numerosity and the perceived numerosity of an illusory pattern. A two-way analysis of variance (ANOVA) for repeated measures (two numerosities, two connectedness levels) confirmed that the pupil response was modulated by both factors (main effect of connectedness: $F(1,15) = 20.5$, $p < 0.001$, $\eta p^2 = 0.6$, $\log_{10}BF = 0.8$; main effect of numerosity: $F(1,15) = 6.2$, $p = 0.025$, $\eta p^2 = 0.3$, $\log_{10}BF = 0.8$), which did not interact ($F(1,15) = 0.13$, $p = 0.72$, $\log_{10}BF = -0.5$).

Figure 3D, E show the pupillary time courses for experiment 2 (using the alternative form of isolated-dot stimuli, with lines removed), which were similar to those of experiment 1. The responses to bright and dark (Fig. 3D), their difference (Fig. 3E) for the four classes of stimuli, and the average difference over the fixed window (1–6 s, Fig. 3F) were all similar to those above them. Again, the difference in pupillary response was strongest for 24

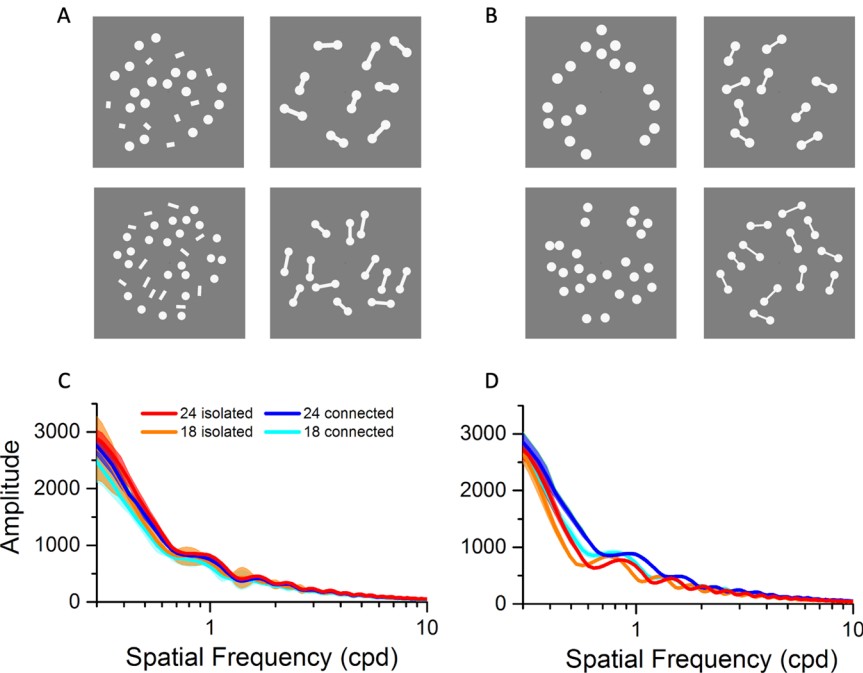

**Fig. 1 Stimuli. A** Examples of white stimuli used for experiment 1: stimuli comprised either 18 or 24 dots, which were either connected by lines or isolated, with the lines displaced to random positions. The stimuli with black dots were identical. Participants simply maintained gaze on a central fixation point without performing any task, while the stimuli were displayed for 6 s. **B** Stimuli for experiment 2. The lines in the connected condition are thinner and in the isolated condition they are absorbed into the dots, which become slightly larger (about 40%). See "Methods" for further details. **C** Fourier amplitude (arbitrary units) of the stimuli of Fig. 1A, as a function of spatial frequency, over the range 0.3–10 cycles per degree. **D** Same as **C**, for the stimuli of Fig. 1B. Source data are provided as a Source Data file.

isolated and weakest for the 18 connected ($t(12) = 2.9$, $p = 0.014$, Cohen's $d = 0.8$, log10BF = 0.6), and intermediate for 24 connected and 18 isolated. Results of the two-way ANOVA for repeated measures were as follows: main effect of connectedness $F(1,12) = 6.7$, $p = 0.024$, $\eta p^2 = 0.4$, log10BF = 0.5; main effect of numerosity $F(1,12) = 2.6$, $p = 0.13$, log10BF = 0; and connectedness by numerosity interaction: $F(1,12) = 0.005$, $p = 0.95$, log10BF = $-0.5$. The reduced effect sizes compared to experiment 1 may be related to the pupillary responses being generally weaker (possibly due to the lower total number of white or dark pixels) and to the illusion being marginally reduced (compare Fig. 2B, D).

**General linear model**. Averaging pupil size over a fixed time window (here 1–6 s) is a standard technique of pupillometry. However, it raises the possibility that the results could depend on the length of the time window. We therefore implemented an additional analysis, modeling the pupil time courses, assuming that they resulted from the linear combination of three predictors convolved with the pupil response function[26]. The pupil response function was estimated for individual participants using their average response time course across conditions (for the parameters of the pupil response function, see Supplementary Table 1). The three predictors were stimulus appearance and disappearance (two impulse-like functions), and time on screen (a boxcar function); their weights represent the strength of the transient onset/offset responses and the sustained pupillary response. Given the estimated pupil response function, we fitted pupil traces from each condition and participant, yielding β-weights representing the contribution of each predictor to the observed pupillary response. Average best fit curves are shown in Fig. 3 (thin lines); their goodness of fit was generally excellent

(83% variance explained averaged across participants and experiments, see Supplementary Table 1).

We focused on the sustained response, considering pupil difference traces (difference between black and white stimuli) to estimate the impact of numerosity on the net pupillary response to luminance. β-Weights for the sustained predictor varied across conditions, reinforcing the results obtained by taking the simple mean of the response over the stimulus window (see Supplementary Tables 2 and 3). β-Weights were highest for 24 isolated dots, lowest for 18 connected dots (experiment 1: $t(15) = 3.8$, $p = 0.002$, Cohen's $d = 0.9$, log10BF = 1.4; experiment 2: $t(12) = 2.2$, $p = 0.047$, Cohen's $d = 0.6$, log10BF = 0.2) and intermediate for 18 isolated or 24 connected dots. For displaced-line stimuli (experiment 1): main effect of connectedness $F(1,15) = 17.1$, $p < 0.001$, $\eta p^2 = 0.5$, log10BF = 0.5; main effect of numerosity $F(1,15) = 3.1$, $p = 0.09$, log10BF = 0.3; connectedness by numerosity interaction: $F(1,15) = 0.1$, $p = 0.79$, log10BF = $-0.5$. For removed-line stimuli (experiment 2): main effect of connectedness $F(1,12) = 3.8$, $p = 0.07$, log10BF = 0.2; main effect of numerosity $F(1,12) = 2.4$, $p = 0.15$, log10BF = $-0.2$; connectedness by numerosity interaction: $F(1,12) = 0.17$, $p = 0.69$, log10BF = $-0.4$.

**Potential artifacts**. Displays with higher perceived numerosity show stronger pupil responses, both dilation to black stimuli and constriction to white stimuli. Thus, any potential artifact that predicts unidirectional pupil changes, constriction or dilation, is unlikely to confound our results.

Although the total number of pixels, and therefore mean luminance, did not vary between conditions in each experiment, the manipulations obviously caused small variations in spatial frequency content, which could have affected the pupillary response[21]. Figure 1C, D plot the spatial frequency amplitude

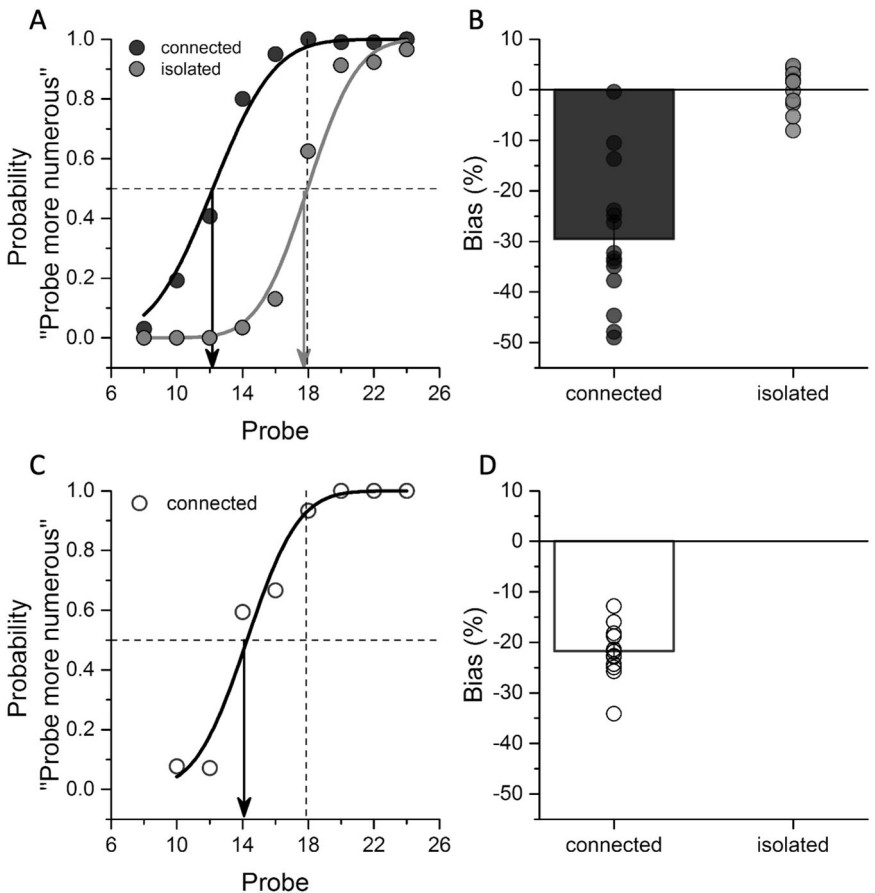

**Fig. 2 Psychophysics results. A** Typical psychometric curves for one participant, measuring the connectedness effect on perceived numerosity for experiment 1. The proportion of trials in which the probe pattern was reported to be more numerous than the standard (which comprised 18 isolated dots) is plotted as a function of the number of dots in the probe pattern. Black and gray lines refer to the connected and isolated conditions, respectively. Arrows indicate the PSE measured in the two conditions. Leftward shifts of the psychometric curve imply underestimation of the test arrays. **B** Perceptual bias expressed as percentage of PSE difference from the reference numerosity for the two connectedness conditions. Bars represent average across participants ($N = 14$ participants, individual data reported as black and gray circles for the connected and isolated conditions, respectively), error bars show 1 SEM. **C, D** Example psychometric curve and average perceptual bias measuring the connectedness effect on perceived numerosity for experiment 2 ($N = 13$ participants, open circles). Bars represent average across participants and error bars show 1 SEM. The results from the first experiment (**A, B**) were replicated in the second one (**C, D**). Source data are provided as a Source Data file.

spectra for the four conditions, separately for the two experiments. Although there are no large differences in amplitude, there are some subtle differences at certain frequencies that could potentially confound the results. However, these small differences are not consistently in the direction needed to explain the effects in both experiments. For example, for experiment 1 (displaced-lines), the amplitude is higher for the isolated conditions, those that elicit stronger pupillary responses. However, the situation reverses for the stimuli of experiment 2 (removed lines), where the amplitude is higher for the connected conditions, associated with a smaller pupillary response. The Fourier amplitude can therefore not explain the results of both experiments, which was a major motivation for using both types of stimulus manipulations.

It is also possible that the eye-movement patterns could have been different for the different classes of stimuli, which would have affected pupillary responses, possibly driving the main effects. We therefore calculated the bivariate confidence interval area for each participant and condition, and analyzed these values with a repeated-measure ANOVA, with connectedness and numerosity levels as factors. None of the main effects or interactions were significant in either of the two pupillometry experiments, suggesting that unstable fixation did not contribute

to the results (see Supplementary Information, Supplementary Fig. 2A, B).

A further possible artifact is that stimuli with higher perceived numerosity have higher perceived brightness, which in turn drives the pupillary response, as has been reported for images of the sun and moon[19]. This seems unlikely, as there are no obvious differences in apparent brightness on inspection of the stimuli of Fig. 1. Nevertheless, we measured perceived brightness in our participants with a forced choice psychophysical technique (see Supplementary Information). The results (Supplementary Fig. 2C) reveal no large, or even significant differences in apparent brightness between the four conditions, excluding the possibility that it is driving the pupillary response.

## Discussion

In this study, we investigated whether pupil size is spontaneously modulated by numerosity, for stimuli of identical luminance. Although participants were not required to judge numerosity, or any other aspect of the stimuli, the magnitude of pupil responses evoked by white or black dots systematically scaled with perceived numerosity: the higher the perceived numerosity, the stronger the pupil response to the stimulus luminance. The pupillary response

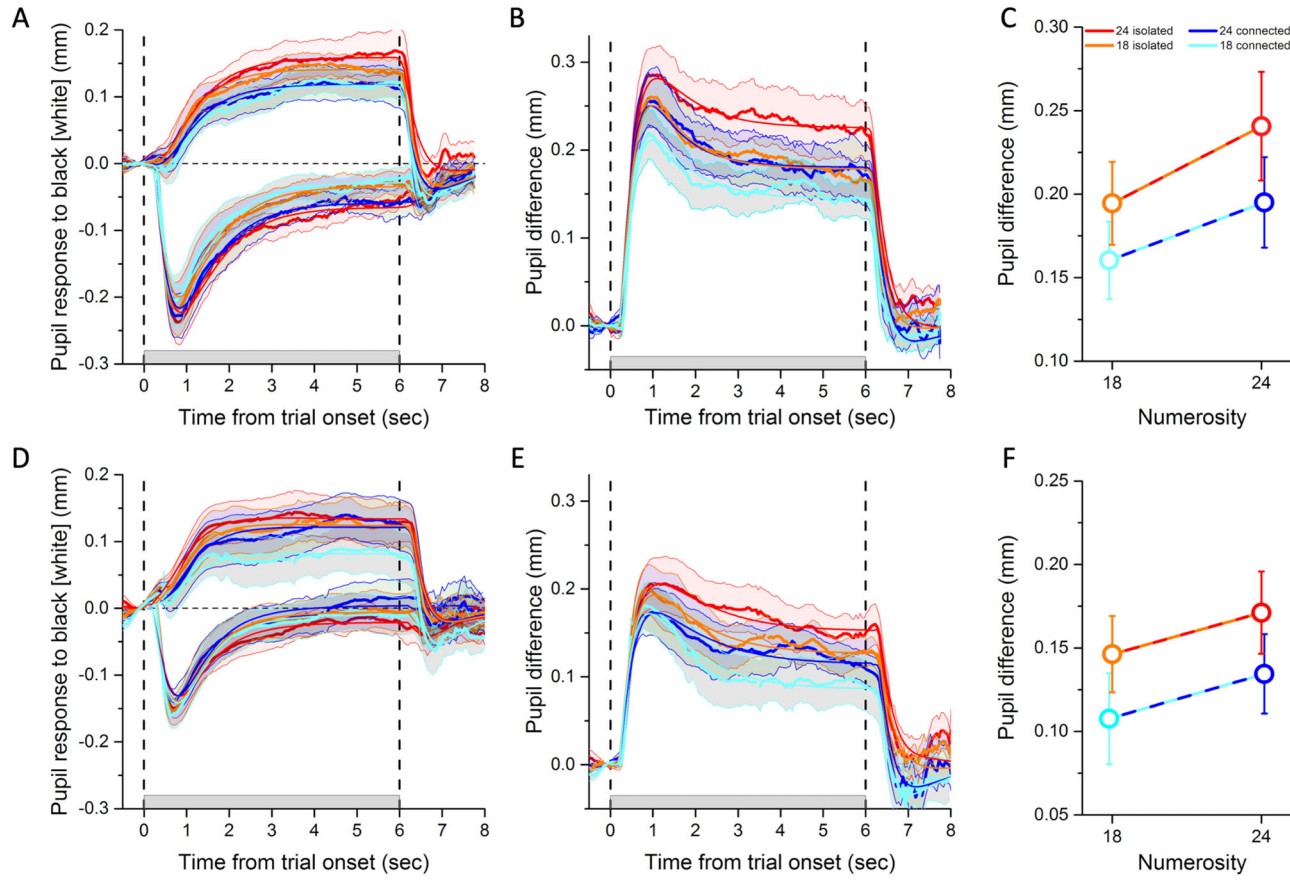

**Fig. 3 Pupil responses to different numerosities and connectedness levels. A** Pupil dilation or constriction in response to black or white stimuli for the displaced-lines condition (experiment 1). Color-coded lines refer to the different numerosity and connectedness levels defined in the legend to **C**. The gray shaded area on the abscissa between the two vertical dashed lines show time of stimulus presentation. Thick lines plot the time courses of the pupil size averaged across participants ($N = 16$ participants) and the shaded area around them the SEM. Thin color-matched lines show the average fit obtained by convolving stimulus predictors with the pupil response function (defined by parameters listed in Supplementary Table 1). **B** Time course of the pupil difference (dark minus light responses) estimating the net pupillary response to luminance. **C** Average pupil difference in the interval 1–6 s after stimulus onset (the stimulus duration minus the first second, which we excluded to discard the fast and transient pupillary onset response). Dots represent averages across participants ($N = 16$ participants) and error bars are SEM. **D–F** Same as **A–C**, for the removed-lines condition (experiment 2). Open circles represent averages across participants ($N = 13$ participants) and error bars are SEM. Repeated-measures ANOVAs tested for pupil differences between numerosity and connectedness levels. Significance of main effects and interactions are reported in text. The results from the first experiment (**A–C**) were replicated in the second one (**D–F**). Source data are provided as a Source Data file.

scaled down when either the physical numerosity was decreased (from 24 to 18) or when the dots were joined by lines to form dumbbells, reducing the perceived numerosity. When the lines joining the dots were displaced or removed, the pupillary response increased. We used two different analyses: quantifying pupil size within a fixed time window (delimited by the stimulus presentation) and fitting the pupil traces on the basis of a physiologically motivated general linear model[26]. Both analyses produced similar results, with both types of stimuli.

The numerosity-driven pupil-size modulation observed in this experiment could not be explained by luminance differences across the stimuli, as the total number of white or black pixels was always exactly matched across all conditions, of both experiments. Nor could the pupillary response be driven by differences in perceived brightness, as has been reported for images of the sun and moon[19]. Psychophysical measurements revealed no statistically significant differences in apparent brightness between the four conditions of our experiment. Although we cannot rule out the possibility that numerosity may affect apparent brightness in some very subtle ways, we can conclude that any potential effects were not robust enough to be driving the responses reported here.

The pupillary response was not driven by differences in the Fourier spectrum, as the spectra were very similar over most of the visible range (Fig. 1C, D) and varied in opposite ways for the stimuli of experiments 1 and 2. Other potential artifacts may include differences in mental effort or memory load. Large differences in these factors are unlikely under the passive viewing conditions of the experiment. In any case, neither of these factors (Fourier power or mental effort) could explain the pattern of the data, as they predict a unidirectional pupil size change: constriction or dilation, respectively[27,28]. This could not account for enhanced constrictor and dilator pupillary response to light and dark stimuli, suggesting multiplicative scaling of the effective stimulus strength.

The results of our study suggest that patterns perceived as more numerous are spontaneously represented as perceptually stronger and consequently evoke stronger pupillary responses. This fits well with previous evidence suggesting that humans spontaneously encode numerosity. Humans are far more sensitive to numerosity changes than to changes in area or density when asked to identify the odd-one-out of three dots arrays without instructions on the aspect the stimuli may differ[29,30]. Similarly,

numerate adults, children, innumerate adults, and monkeys used numerosity rather than other non-numerical dimensions to classify arrays of dots as "little" or "a lot"[31]. The current results are also in line with other studies suggesting that numerosity is a salient visual feature that is difficult to ignore[32], and can drive automatic oculomotor orientation responses[12].

The mechanisms by which perceived numerosity enhances the pupillary response are far from clear. One possibility is that the effect is mediated by attention, given that pupillary responses are known to be enhanced when attention is directed towards stimuli of different luminance[15,16]. This is certainly feasible, although it is not obvious why participants should spontaneously attend to higher perceived numerosities and maintain attention for the whole 6 s period. Whatever the underlying mechanism, the results show that stimuli of higher perceived numerosity have a higher salience, or stimulus strength, and this is reflected in the gain of the pupillary response.

The pupillary response to luminance is one of the simplest sensory responses. Pupil control is completely involuntary[33], with luminance regulation mediated by a simple subcortical circuit that starts from the retina and sends light flux information to the pupillomotor Edinger-Westphal (EW) nuclei of the tectum via the olivary pretectal nucleus (OPN)[14]. Clearly, the modulation of luminance responses by perceptual and cognitive factors, such as brightness and size illusions, or attention shifts[15–18], implies cortical modulation of this circuit. A recent neurophysiological study in monkeys showed that electrical micro-stimulation of the prefrontal cortex (specifically of the frontal eye field), which is implicated in attentional control, modulated the pupillary light reflex in a spatial- and temporal-specific manner[34]. This result suggests that the prefrontal cortex may exert control over the mesencephalic pupil light reflex circuit (OPN and EW), either through direct feedback signals or indirectly via relay stations in the occipital cortex and/or in the superior colliculus[22]. It is well documented that prefrontal and occipito-parietal structures support numerosity processing[35–37], even in passive viewing paradigms[38,39], and there is also evidence that numerosity may be already coded at subcortical levels[12,40]. Moreover, electro-encephalogram and functional magnetic resonance imaging studies have shown that the effect of connectedness on numerosity perception emerges as early as 150 ms after stimulus onset, at the level of V3[41], and continues in the parietal cortex[42]. These cortical and subcortical connections could support the spontaneous modulation of the pupillary response to luminance with perceived numerosity.

In conclusion, our results reinforce previous work suggesting that numerosity is a primary visual attribute, which spontaneously modulates one of the more basic sensory responses, the pupil light response. Pupillometry may prove to be an effective tool to study numerical cognition, providing a quantitative and objective index that tracks this perceptual process. The paradigm is simple, requires no specific training, is task free, and responses are recorded automatically without the need for invasive experimenter intervention. These features make it a potential candidate for future studies on populations and species for which psychophysical testing has proved difficult or unfeasible.

## Methods
**Participants.** Sixteen participants (six males, mean age: 30 ± 3 years) with normal or corrected to normal vision participated in the study. All participants took part to the first pupillometry experiment, 14 of these participated in the numerosity discrimination experiment of Fig. 2, and 13 in the second pupillometry and psychophysics experiment. The research was approved by the local ethics committee (Commissione per l'Etica della Ricerca, University of Florence, n. 111 dated 7 July 2020) and was in accordance with the Declaration of Helsinki. All participants gave written informed consent prior to the study.

**Stimuli and apparatus.** For each experiment, 8 types of stimuli were used, illustrated in Fig. 1: 2 numerosities (18 and 24), connected into dumbbell-like shapes or isolated (either by displacing or removing the connecting lines). All stimulus types could be either black or white (maximum and minimum luminances on the screen, about 12.6 and 256 cd/m²), presented on a gray background of 129.3 cd/m². We ran two separate experiments, with slightly different stimuli and hence different Fourier spectra (Fig. 1), to confront several potential artifacts. In both experiments, all stimuli had the same number of pixels, covering 92.7 deg² for experiment 1 and 82.3 deg² for experiment 2. For experiment 1 (Fig. 1A), the connecting lines were displaced to random positions on the screen in the isolated-dot condition: dot diameter, 2.2° for N18, 1.9° for N24; line width, 1.03°; line length 2°–3°. For experiment 2 (Fig. 1B), the connecting lines were removed in the isolated condition and their pixels absorbed into the dots: dot diameter, 2.2° for N18, 1.9° for N24 in connected condition; 2.44°–2.64° (N18) and 2.10°–2.28° (N24) in isolated condition; line width, 0.6°; line length, 1.6°–2.8°.

To match the total number of pixels in the arrays (and hence luminance), as well as the covered area (convex hull), stimuli were precalculated offline through multiple steps. In the first step, the isolated stimuli were created: coordinates were randomly generated for twice the number of the desired dot positions, half for the dots and the other half for the lines, with the constraint that neither dots nor lines could be closer than 0.5°. In a second step, the connected stimuli were created by connecting randomly chosen couples of dots, with line length as detailed above. In the third step, the total number of displayed pixels and the convex hull of the isolated dots was modified to match those of the connected stimuli. For the stimuli of experiment 2, dots were enlarged so that the overall ink of the isolated stimulus matched those of the dumbbells (which contained the connectors). For stimuli of experiment 1, the length of the isolated lines was iteratively adjusted until it matched the total number of pixels of the connected stimuli (which would otherwise contain less pixels due to the over imposition of the lines with the dots at the point of connection between the two). Then the position of the isolated dot pairs was iteratively modified until the convex hull matched that of the connected stimuli (which would otherwise have had higher probability of appearing sparser than the isolated stimuli). The final convex hull was 513 deg².

Connected and isolated dots were intermixed within a session, whereas arrays of different numerosity or color (white or black) were presented in separate sessions. Order of trials and sessions was varied pseudo-randomly across participants. Each participant performed 4 sessions of 60 trials each. Trials started with a fixation point shown for 1 s, followed by the presentation of one of the numerosity arrays, which remained visible for 6 s. Trials ended 1 s after stimulus disappearance, giving a 2 s interstimulus interval. Importantly, participants were instructed to keep their gaze on the fixation point and simply observe the stimuli, without performing any task.

Measurements were made in a quiet dark room with participants sitting in an experimental booth surrounded by thick black curtains so that the only light source was the stimulus display (Liquid Crystal Display monitor screen, 1280 × 720 pixels, refresh rate 60 Hz). Participants sat at 57 cm from the screen, with head stabilized by chin rest. Pupil diameter was monitored at 500 Hz with an EyeLink 1000 system (SR research) with infrared camera mounted below the screen, recording from the left eye. Before each session, eye position was linearized by a standard nine-point calibration routine. Stimuli were generated and presented under Matlab using Psychtoolbox-3 routines[43].

**Psychophysics.** A subset of participants took part in psychophysical experiments to establish the strength of the illusion (after completing the first set of pupillometry measurements and usually on a different day) and a brightness comparison task as well (see Supplementary Information). For the numerosity measurements, two arrays of dots (test and reference stimuli) were presented sequentially in a randomized order for 500 ms in central fixation and participants judged which was the more numerous. In separate sessions, stimuli were either all black or all white, in analogy with the pupillometry experiment. The test stimulus of experiment 1 comprised isolated dots and lines, of variable numerosity (8, 10, 12, 14, 16, 20, 22, or 24 dots, with half the number of lines), whereas the test stimulus of experiment 2 comprised only isolated dots, of variable numerosity (spanning from 5 to 12 dots). The reference stimulus (randomly presented first or second) had a fixed number of 18 dots and 9 lines, which could be either isolated or connected for experiment 1, whereas only the connected condition was tested for experiment 2. For each condition (color and connectedness), participants were tested with 3 sessions of 40 trials each. The participant was asked to report whether the first or second array had more dots, by pressing the corresponding key.

**Data analysis.** Pupillometry data were preprocessed to exclude blinks or signal losses. Specifically, we excluded time points where pupil size was unrealistically small (< 0.1 mm), where pupil size changes where unrealistically large (> 1 mm from the median of the trial), or too quick (any pupil size changes that were faster than 25 mm/s were treated as artifacts and the surrounding 20 ms window were excluded from the analyses). Data points that passed this quality check were down-sampled at 20 Hz and then high-pass filtered by convolving the trace with a 500 ms square window. The resulting pupil size time courses were baseline-corrected by subtracting the average pupil diameter in the 200 ms preceding the stimulus onset. To statistically compare pupil size changes across conditions, we averaged the pupil

size over a period of 1–6 s after the stimulus onset, i.e., over the stimulus presentation window excluding the first second (which included the pupil size fast response triggered by the stimulus appearance). These values were analyzed with two-way repeated-measures ANOVA, with numerosity and connectedness level as factors.

We additionally modeled the pupil time courses with a General Linear Model approach, assuming that pupil responses resulted from the linear combination of three predictors (two impulse-like and one boxcar functions, each normalized to its integral) convolved by the pupil response function. The three predictors corresponded to three components known to contribute to pupil size[21]: a transient pupil modulation at the onset and offset of the stimulus and a sustained response lasting for the entire stimulus presentation time window. It has been proposed that these components are mediated by different mechanisms and have different characteristics: the transient pathway is characterized by poor spatial summation, bandpass temporal response, and high-contrast gain, whereas the sustained pathway exhibits large spatial summation, low-pass temporal response, and low-contrast gain[21]. The transient predictors (impulse-like functions at stimulus onset and offset) were to capture the grating response, which has previously been reported to be generated by visual transients[21,28]. The sustained predictor (a boxcar function representing stimulus duration) captures a steady-state luminance response, which is primarily driven by illumination but can be affected by perceptual and cognitive factors[44].

The pupil response function was modeled by a Gamma function $h(t)$:

$$h(t) = \frac{(\frac{t-\delta}{\tau})^{(n-1)} e^{-(\frac{t-\delta}{\tau})}}{\tau(n-1)!} \quad (1)$$

where $n$ was the number of filters, $\tau$ the change rate, and $\delta$ the delay in pupil response after stimulus onset. For each participant we found the best-fitting parameters $n$, $\tau$, and $\delta$ (each constrained in an appropriate interval: from 1 to 8, 10 to 800, and 0 to 200, respectively, see Supplementary Table 1). Once we defined the pupil response function that best fitted the average pupil trace across conditions, we estimated the β-weights for each condition separately and computed the goodness of fit of each of them. We then statistically compared the β-weights across conditions by entering them in two-way repeated-measure ANOVA with numerosity and connectedness level as factors.

To analyze the results of the numerosity discrimination task, for each condition (black or white arrays, connected or isolated), the responses were plotted as function of the probe numerosity and fitted with a cumulative Gaussian distribution, whose median defines the PSE (see Fig. 2C). We then computed bias index as:

$$Bias = 100 * \left(\frac{PSE}{N} - 1\right) \quad (2)$$

Statistical tests (ANOVA, $t$-tests, and Bayesian analyses) were conducted with Matlab or Jasp 0.14.1[45]. For Bayesian ANOVA, models including the connectedness and numerosity factors, both these factors, and both factors plus the interaction were ordered by their predictive performance relative to the best model. The reported Bayes factors correspond to inclusion Bayes factors resulting from the analysis of the effects across all matched models[46]. Bayes factors are reported in logarithmic base 10 units (log10BF) and their absolute values should be interpreted as providing anecdotal (0–0.5), substantial (0.5–1), strong (1–1.5), or very strong (>1.5) evidence, in favor of the alternative hypothesis if positive or the null hypothesis if negative.

**Reporting summary**. Further information on research design is available in the Nature Research Reporting Summary linked to this article.

## Data availability
Data are available at Zenodo (https://doi.org/10.5281/zenodo.5168707)[47]. Source data are provided with this paper.

## Code availability
Codes and scripts will be provided upon reasonable request from the corresponding author.

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

## Acknowledgements

This research has received funding from the European Union's Horizon 2020 research and innovation program under the Marie Skłodowska-Curie (grant agreement number 885672–DYSC-EYE-7T "The neural substrate of numerical cognition in dyscalculia revealed by eye tracking and ultra-high field 7T functional magnetic imaging") and from the European Research Council (ERC) under the European Union's Horizon 2020 research and innovation programs (grant agreement number 801715–PUPILTRAITS and number 832813–GenPercept).

## Author contributions

All authors contributed to the study concept and to the design. Stimuli were designed by G.M.C., E.C., and A.P. Testing and data collection were performed by E.C. and A.P. E.C., A.P., and P.B. performed the data analysis. All authors contributed to the interpretation of results. E.C. drafted the manuscript, and D.B. and P.B. provided critical revisions. All authors approved the final version of the manuscript for submission.

## Competing interests

The authors declare no competing interests.
