## [Peer Review File · Nature Communications]

The pupil responds spontaneously to perceived numerosityREVIEWER COMMENTS

Reviewer #1 (Remarks to the Author):

Castaldi and colleagues record pupil sizes in observers who are presented with stimuli of varying numerosity (18 or 24 dots, either connected or unconnected) and find that displays perceived as being of higher numerosity yield a stronger pupil effect (more dilation for dark, more constriction for bright stimuli). From the speed and robustness of the effect, they conclude that numerosity is a primary visual attribute.

The experiment is well-conducted and the analysis sound (provided the 5s interval was chosen a priori, which I assume – the time resolved analysis of figure 3 is useful in any case). In particular, I appreciate the care given to the stimulus design that avoids confounding numerosity with various other visual attributes, in particular luminance. I have, however, two conceptual concerns, the first of which might require an additional control experiment, while the second should be readily addressable with the data at hand:

(1) I understand that stimuli are matched for luminance, but does this hold for brightness perception? That is, can the authors exclude that stimuli that are perceived as more numerous are also perceived as brighter in case of white dots and as darker in case of black dots? This could (and in my mind should) be tested psychophysically in a new group of observers. If this would be the case, the results would still be interesting, but the interpretation of numerosity as “primary visual attribute” would be less clear (the effect could then be mediated by the perception of brightness).

(2) Observers were instructed to fixate. Did the authors check that there were no differences in eye-movement patterns (including fixational eye movements to the extent the Eyelink can measure them) between the stimuli? As this is potential confound for the pupil measurements, this should be verified.

Reviewer #2 (Remarks to the Author):

The idea of testing automatic and rapid processing of numerosity with pupillometry strikes me as a novel and interesting idea. Such a finding is important for understanding a basic cognitive mechanism that we also share with other animals.

The effects are clear and well documented by thorough analyses. Several aspects of the experimental design (i.e., using both dark and bright stimuli, controlling luminance and array size) allow ruling out several possible accounts of the findings. The fact that pupillary changes, regardless of their directions (i.e., either as relative dilations or constrictions), were similarly enhanced indicate as the underlying mechanism something else than - for example - ‘mental effort.’ If so, one would expect that the more spatial locations there are to process, the more mental effort should be allocated to the task, which (lines 215-217) should have caused pupillary changes in one direction (dilations).

The authors suggest that the underlying mechanism causing these systematic pupillary changes is ‘perceptual strength.’ That is, “patterns perceived as more numerous are spontaneously represented as perceptually stronger.” The strength of the percept is mirrored in the strength of the pupillary response. Just like a stronger light causes a stronger constriction, perceiving more ‘lights’ (or more luminous items, but not more bright pixels) leads to stronger constrictions (or conversely dilations for dark items).

The account proposed by the authors seems reasonable and interesting from a psychological view. However, in my opinion, the authors dismiss too quickly the possibility that some other, intermediate, mechanisms may also be at work, in particular attention. Although the authors do not rule out the possibility that attention mediates the response strength of the pupil (in line with evidence from previous studies, Ref. 10), they express skepticism for essentially three reasons. However, in my opinion, each of the three reasons they invoked seem to me rather unconvincing. The first suggested reason is that because all stimuli were centered and shown alone, without competing stimuli or

distractors, “it is difficult to imagine that participants would be attending differentially to stimuli depending on their numerosity.” It seems to me that if paying attention to numerosity is important to survival, then one should imagine exactly some differential attentional engagement based on numerosity. This could be derived from the initial statement in the paper that “being able to efficiently estimate the number of enemies or prey is essential for survival: all animals – from insects to humans - are capable of some form of numerosity discrimination.” The second reason mentioned is that the numerosity effect was sustained for several seconds in the pupil waveform. The authors find “difficult to imagine why attention would be engaged so systematically over such a long interval on a stimulus that is not even task relevant.” Again, it seems strange to describe the stimulus as irrelevant, since the task for participants is explicitly to look (i.e., overtly attend) at each stimulus. Moreover, it seems difficult for me to imagine that participants would not be able to attend to each display – especially in the absence of anything else to look at - for just a few seconds. Similarly, the fact that a task was ‘passive’ (i.e., no response was required) did not make necessarily exempt of attention the task of passively observing. The third reason, stressed by the authors as “most important,” is that “there is no a priori reason to predict that attention should be biased towards more numerous arrays. In fact, it is easy to imagine situations where less numerous arrays are more salient, such as at a supermarket, where the less numerous line at the cashiers may draw our attention.” Based on their own theoretical framework, if numerosity is biologically relevant to the organism, more numerous patterns should capture attention relatively more than less numerous ones. Their counterexample about queues in a supermarket is not a convincing argument, being unrelated from the biological or evolutionary motivation for a privileged numerosity processing. I advise the authors to drop the supermarket scenario from their argument.

Hence, I suggest that either the authors propose novel arguments against the idea of an attentional mediation or consider this as a viable alternative to automatic and spontaneous perceptual encoding.

One more concern I think the authors could address is ‘density’ or ‘spatial frequency’ information in the display. These, at least by looking at Figure 1A, could differ between the connected and unconnected items, since the “connecting lines” in the latter displays appear to add density to the stimuli. In fact, if there were more elements in some patterns but the size of the array (line 21: convex hull) was kept constant, density and spatial frequency should increase. Previous work by Barbur (e.g., Ref. 15) has shown that spatial frequency affects pupil size in equiluminant stimuli. The authors could check and perhaps dispel the possibility that some of the observed difference are due to uncontrolled structural differences in the arrays.

Figure 3 is not necessary. It does not seem to reveal additional effects than those already visible in Figure 2 (e.g., the rapid timing of responses) and, in fact, it mainly depicts event that fail to reach statistical significance.

Finally, I am not sure the Title (‘The pupil responds spontaneously to perceived number of items’) reflects appropriately the findings, since what the study tests is ‘numerosity’ and not the ability to identify ‘the number of items’ (i.e., any actual count). As the authors know well, numerosity is the basic ability to distinguish the some mass contains more or less of something than another comparison mass (i.e., and ‘ordinal’ mental measurement) without necessarily knowing how many (‘the number of’) items compose each mass (i.e., a ‘ratio’ mental measurement).

Reviewer #3 (Remarks to the Author):

This study tested pupillary responses to dot arrays that are known to induce numerosity-related illusions. It is well known that, when dot arrays are connected in pairs by a thin line, the numerosity of an array is reliably underestimated compared to the numerosity of the exact same array without the lines connecting dots in pairs. The authors tested whether higher-level brain areas modulate pupil size to reflect perceived numerosity or true numerosity. The results show that pupillary constriction and dilation were modulated by both true and perceived numerosity (in the authors terms, the main effect of numerosity and the main effect of connectedness, respectively).

This is an ingenious study with clear competing hypotheses and meaningful results. The work is timely, and the paper is written well. I do, however, have one major concern along with several other issues with some suggestions for improvement.

In order to equate the overall luminance (or pixels), the authors designed the “isolated” arrays to include all the lines but unconnected to any of the dots. However, the lines look fairly thick and being intermixed with all the dots, the “isolated” array actually looks like a big cluster of heterogeneous items. On the other hand, the “connected” array looks like a cluster of larger barbells. In other words, the two types of arrays (isolated vs. connected) look qualitatively different. They do not look like the same dot array but with and without lines. I am surprised to see why they have chosen to include all the freely floating lines in the isolated arrays. By doing so, it makes the results unclear if they demonstrate the effect of illusory underestimation or just the effect of numerosity of random, heterogeneously sized items, or some other idiosyncratic interaction between the two effects. It would have been better had they used other approaches such as attaching halved segments of the lines to the dots (Franconeri, Bemis, & Alvarez, 2009, *Cognition*) or not having the lines at all (Fornaciai, Cicchini, & Burr, 2009, *Cognition*) for the isolated array. The psychophysical experiment does not help much here either. Unless I’m missing something in the manuscript, it’s not clear exactly what kind of instructions were given to the participants. Were they asked to make estimations just on the dots, ignoring all the lines connected or not? In any case, if the lines are so thick and as big as some of the dots, it would be difficult for participants to discern which are the dots that they need to attend to and which are “distractors.” It is not too surprising that participants overestimate isolated arrays which contain randomly looking heterogeneous items.

I suggest that the results from the GLM incorporating pupil response function (text in p. 8) to appear immediately after the results from the overall mean pupil differences across the whole interval (i.e., at the end of p. 6), followed by the results from the time-point by time-point analysis. Analysis on the raw mean pupil differences is much more logically related to the analysis based on the GLM. Also, the time-point by time-point analysis is very important but the data are too noisy, which makes some of the time marks not too convincing. I would either perform some appropriate filtering or perform the time-point analysis on the beta weights from the GLM.

The experiment has a nice 2x2x2 design with luminance (black vs. white), numerosity (18 vs. 24), and connectedness (connected vs. not) as factors. I understand that the effect of luminance was collapsed in some cases (e.g., Fig. 2), but at least the other two orthogonal factors make it possible to interpret all the results in terms of the effects of numerosity, connectedness, and the interaction. Although this was done in some places, the results are not displayed as such in many other places – for example, in discussing the ANOVA on the beta weights from the GLM. In the results of the time-point analysis (Fig. 3), I understand the first three lines represent each of the main effects in the 2x2x2 design, but “18 connected vs. 24 isolated” seems out of left field. It would be more important to plot the temporal pattern of the interaction effect, instead.

Other suggestions and comments:

The Introduction brings attention to previous work on neural mechanisms of numerosity perception. The examples of single neurons or topographic organization of numerosity representation are relevant, but they do not directly speak to the nature of early/fast sensory processing, which is central to the study. More relevant findings may be demonstrations of a rapid encoding of numerosity in the cortical visual stream (e.g., Park, DeWind, Woldorff, & Brannon, 2016, *Cerebral Cortex*).

p. 7 “only twice the latency of the luminance response” – The word “only” here implies more than what the data should imply. It is dangerous to assume that a mental process occurs strictly in an ordinal scale. The authors probably do not mean this, but it sounds as if mental processes that can happen within 600 ms are twice as much as mental processes that can happen within 300 ms.

p. 9 The sentence starts with “Most convincingly” should be unpacked for a clear understanding.

In Methods, please describe what the participants were told to do.

Please use $\log_{10}BF$ instead of $IgBF$.

General

We sincerely thank all three reviewers, both for your generally positive appraisal of our study, and for the very helpful suggestions for improving the study and the manuscript.

As you will see, we took your concerns very seriously, addressing them with additional experiments and analyses. We ran a completely new experiment on most participants, confirming the pupillometry effect with different stimuli that did not have the potential artifact mentioned by referee 3. These extra data also address the possible artifact of different spatial frequency content of the stimuli, mentioned by referee 2, as their Fourier content is opposite to that of the other stimuli. We also measured the apparent brightness of the stimuli and eye-movements as requested by referee 1, showing these were unlikely to be driving the pupillometry effect.

We believe that these extra data and analysis, together with a thorough revision of the manuscript, address all the concerns raised: as well as providing an internal replication of the main result.

Below we give detailed replies to all the comments. Major changes in the manuscript are marked in blue.

Specific

Reviewer #1 (Remarks to the Author):

Castaldi and colleagues record pupil sizes in observers who are presented with stimuli of varying numerosity (18 or 24 dots, either connected or unconnected) and find that displays perceived as being of higher numerosity yield a stronger pupil effect (more dilation for dark, more constriction for bright stimuli). From the speed and robustness of the effect, they conclude that numerosity is a primary visual attribute.

The experiment is well-conducted and the analysis sound (provided the 5s interval was chosen a priori, which I assume – the time resolved analysis of figure 3 is useful in any case). In particular, I appreciate the care given to the stimulus design that avoids confounding numerosity with various other visual attributes, in particular luminance. I have, however, two conceptual concerns, the first of which might require an additional control experiment, while the second should be readily addressable with the data at hand:

Thank you for the positive evaluation of our work. On the advice of Referee 2, we have decided to remove the time-resolved analysis, but point out that we also tested the effect with a GLM, which did not presume any particular time window.

(1) I understand that stimuli are matched for luminance, but does this hold for brightness perception? That is, can the authors exclude that stimuli that are perceived as more numerous are also perceived as brighter in case of white dots and as darker in case of black dots? This could (and in my mind should) be tested psychophysically in a new group of observers. If this would be the case, the results

would still be interesting, but the interpretation of numerosity as “primary visual attribute” would be less clear (the effect could then be mediated by the perception of brightness).

This is a very interesting point, which we have met head on by measuring apparent luminance of the stimuli. We now include a supplementary figure (Figure S1C) showing that we could find no significant differences in apparent brightness of the four classes of stimuli used in the main experiment. This is consistent with there being no brightness differences visible on inspection of the stimuli in Figures 1.

However, we do like the idea, for which we thank the reviewer, and are reluctant to dismiss entirely the possibility that numerosity causes subtle effects on brightness, paralleling those of the pupil; but revealing these small effects would require very subtle psychophysical techniques, beyond the scope of this experiment.

(2) Observers were instructed to fixate. Did the authors check that there were no differences in eye-movement patterns (including fixational eye movements to the extend the Eyelink can measure them) between the stimuli? As this is potential confound for the pupil measurements, this should be verified.

We have performed a detailed analysis of eye-movements in the various conditions (new Figure S1 A-B), and show they cannot explain the pupil difference between conditions. We now report the eye-movement analysis in supplementary material.

Reviewer #2 (Remarks to the Author):

The idea of testing automatic and rapid processing of numerosity with pupillometry strikes me as a novel and interesting idea. Such a finding is important for understanding a basic cognitive mechanism that we also share with other animals.

The effects are clear and well documented by thorough analyses. Several aspects of the experimental design (i.e., using both dark and bright stimuli, controlling luminance and array size) allow ruling out several possible accounts of the findings. The fact that pupillary changes, regardless of their directions (i.e., either as relative dilations or constrictions), were similarly enhanced indicate as the underlying mechanism something else than - for example - ‘mental effort.’ If so, one would expect that the more spatial locations there are to process, the more mental effort should be allocated to the task, which (lines 215-217) should have caused pupillary changes in one direction (dilations).

Thank you for the positive evaluation of our work.

The authors suggest that the underlying mechanism causing these systematic pupillary changes is ‘perceptual strength.’ That is, “patterns perceived as more numerous are spontaneously represented as perceptually stronger.” The strength of the percept is mirrored in the strength of the pupillary response. Just like a stronger light causes a stronger constriction, perceiving more ‘lights’ (or more luminous items, but not more bright pixels) leads to stronger constrictions (or conversely dilations for dark items).

The account proposed by the authors seems reasonable and interesting from a psychological view. However, in my opinion, the authors dismiss too quickly the possibility that some other, intermediate, mechanisms may also be at work, in particular attention. Although the authors do not rule out the possibility that attention mediates the response strength of the pupil (in line with evidence from previous studies, Ref. 10), they express skepticism for essentially three reasons. However, in my opinion, each of the three reasons they invoked seem to me rather unconvincing. The first suggested reason is that because all stimuli were centered and shown alone, without competing stimuli or distractors, “it is difficult to imagine that participants would be attending differentially to stimuli depending on their numerosity.” It seems to me that if paying attention to numerosity is important to survival, then one should imagine exactly some differential attentional engagement based on numerosity. This could be derived from the initial statement in the paper that “being able to efficiently estimate the number of enemies or prey is essential for survival: all animals – from insects to humans - are capable of some form of numerosity discrimination.” The second reason mentioned is that the numerosity effect was sustained for several seconds in the pupil waveform. The authors find “difficult to imagine why attention would be engaged so systematically over such a long interval on a stimulus that is not even task relevant.” Again, it seems strange to describe the stimulus as irrelevant, since the task for participants is explicitly to look (i.e., overtly attend) at each stimulus. Moreover, it seems difficult for me to imagine that participants would not be able to attend to each display – especially in the absence of anything else to look at - for just a few seconds. Similarly, the fact that a task was ‘passive’ (i.e., no response was required) did not make necessarily exempt of attention the task of passively observing. The third reason, stressed by the authors as “most important,” is that “there is no a priori reason to predict that attention should be biased towards more numerous arrays. In fact, it is easy to imagine situations where less numerous arrays are more salient, such as at a supermarket, where the less numerous line at the cashiers may draw our attention.” Based on their own theoretical framework, if numerosity is biologically relevant to the organism, more numerous patterns should capture attention relatively more than less numerous ones. Their counterexample about queues in a supermarket is not a convincing argument, being unrelated from the biological or evolutionary motivation for a privileged numerosity processing. I advise the authors to drop the supermarket scenario from their argument.

Hence, I suggest that either the authors propose novel arguments against the idea of an attentional mediation or consider this as a viable alternative to automatic and spontaneous perceptual encoding.

We have to admit, you are right! Our arguments were weak, and we have now toned them down completely, acknowledging that attention may be involved. At this stage we think it unlikely: but if it were to be proven to be implicated in some way, it would not reduce the impact of our study.

One more concern I think the authors could address is ‘density’ or ‘spatial frequency’ information in the display. These, at least by looking at Figure 1A, could differ between the connected and unconnected items, since the “connecting lines” in the latter displays appear to add density to the stimuli. In fact, if there were more elements in some patterns but the size of the array (line 21: convex hull) was kept constant, density and spatial frequency should increase. Previous work by Barbur

(e.g., Ref. 15) has shown that spatial frequency affects pupil size in equiluminant stimuli. The authors could check and perhaps dispel the possibility that some of the observed differences are due to uncontrolled structural differences in the arrays.

This is a good point, which we believe we have resolved with the additional experiment requested by Referee 3. We now add Fourier Transforms of all the stimuli used. As you can see, they are very similar for all classes of stimuli. Importantly, the small variations in Fourier amplitude are in the opposite direction for the new stimuli (no lines in the isolated condition), allowing us to rule out this potential artifact. Thank you.

Figure 3 is not necessary. It does not seem to reveal additional effects than those already visible in Figure 2 (e.g., the rapid timing of responses) and, in fact, it mainly depicts events that fail to reach statistical significance.

We agree with you, and it may be misleading, so we have removed the figure. Thank you.

Finally, I am not sure the Title ('The pupil responds spontaneously to perceived number of items') reflects appropriately the findings, since what the study tests is 'numerosity' and not the ability to identify 'the number of items' (i.e., any actual count). As the authors know well, numerosity is the basic ability to distinguish the some mass contains more or less of something than another comparison mass (i.e., and 'ordinal' mental measurement) without necessarily knowing how many ('the number of') items compose each mass (i.e., a 'ratio' mental measurement).

Thanks, we agree completely and have changed the title as suggested.

Reviewer #3 (Remarks to the Author):

This study tested pupillary responses to dot arrays that are known to induce numerosity-related illusions. It is well known that, when dot arrays are connected in pairs by a thin line, the numerosity of an array is reliably underestimated compared to the numerosity of the exact same array without the lines connecting dots in pairs. The authors tested whether higher-level brain areas modulate pupil size to reflect perceived numerosity or true numerosity. The results show that pupillary constriction and dilation were modulated by both true and perceived numerosity (in the authors' terms, the main effect of numerosity and the main effect of connectedness, respectively).

This is an ingenious study with clear competing hypotheses and meaningful results. The work is timely, and the paper is written well. I do, however, have one major concern along with several other issues with some suggestions for improvement.

Thank you.

In order to equate the overall luminance (or pixels), the authors designed the "isolated" arrays to include all the lines but unconnected to any of the dots. However, the lines look fairly thick and being intermixed with all the dots, the "isolated" array

actually looks like a big cluster of heterogeneous items. On the other hand, the “connected” array looks like a cluster of larger barbells. In other words, the two types of arrays (isolated vs. connected) look qualitatively different. They do not look like the same dot array but with and without lines. I am surprised to see why they have chosen to include all the freely floating lines in the isolated arrays. By doing so, it makes the results unclear if they demonstrate the effect of illusory underestimation or just the effect of numerosity of random, heterogeneously sized items, or some other idiosyncratic interaction between the two effects. It would have been better had they used other approaches such as attaching halved segments of the lines to the dots (Franconeri, Bemis, & Alvarez, 2009, Cognition) or not having the lines at all (Fornaciai, Cicchini, & Burr, 2009, Cognition) for the isolated array. The psychophysical experiment does not help much here either. Unless I’m missing something in the manuscript, it’s not clear exactly what kind of instructions were given to the participants. Were they asked to make estimations just on the dots, ignoring all the lines connected or not? In any case, if the lines are so thick and as big as some of the dots, it would be difficult for participants to discern which are the dots that they need to attend to and which are “distractors.” It is not too surprising that participants overestimate isolated arrays which contain randomly looking heterogeneous items.

Thank you for pointing this out. We made the lines thick to provide more bright or dark pixels to drive the pupil response, but agree this is a potential confound. We have therefore conducted a new experiment using control stimuli with thinner lines in the connected condition and no lines in the isolated condition (pixels absorbed into the disks). The effects were replicated, addressing this important potential concern, as well as that raised by Referee 2, while providing an internal replication.

I suggest that the results from the GLM incorporating pupil response function (text in p. 8) to appear immediate after the results from the overall mean pupil differences across the whole interval (i.e., at the end of p. 6), followed by the results from the time-point by time-point analysis. Analysis on the raw mean pupil differences is much more logically related to the analysis based on the GLM. Also, the time-point by time-point analysis is very important but the data are too noisy, which makes some of the time marks not too convincing. I would either perform some appropriate filtering or perform the time-point analysis on the beta weights from the GLM.

We agree, but considering the comments of referee 2, we have decided to drop the old Figure 3, it could have been misleading.

The experiment has a nice 2x2x2 design with luminance (black vs. white), numerosity (18 vs. 24), and connectedness (connected vs. not) as factors. I understand that the effect of luminance was collapsed in some cases (e.g., Fig. 2), but at least the other two orthogonal factors make it possible to interpret all the results in terms of the effects of numerosity, connectedness, and the interaction. Although this was done in some places, the results are not displayed as such in many other places – for example, in discussing the ANOVA on the beta weights from the GLM. In the results of the time-point analysis (Fig. 3), I understand the first three lines represent each of the main effects in the 2x2x2 design, but “18 connected vs. 24 isolated” seems out of left field. It would be more important to plot the temporal pattern of the interaction effect, instead.

Thank you. We have changed our bar plots to a 2 X 2 plot as suggested, and agree it works much better. We also now report the interaction where omitted (always not significant) and, as mentioned above, dropped figure 3.

Other suggestions and comments:

The Introduction brings attention to previous work on neural mechanisms of numerosity perception. The examples of single neurons or topographic organization of numerosity representation are relevant, but they do not directly speak to the nature of early/fast sensory processing, which is central to the study. More relevant findings may be demonstrations of a rapid encoding of numerosity in the cortical visual stream (e.g., Park, DeWind, Woldorff, & Brannon, 2016, Cerebral Cortex).

Thank you, now referenced.

p. 7 “only twice the latency of the luminance response” – The word “only” here implies more than what the data should imply. It is dangerous to assume that a mental process occurs strictly in an ordinal scale. The authors probably do not mean this, but it sounds as if mental processes that can happen within 600 ms are twice as much as mental processes that can happen within 300 ms.

Yes, we certainly did NOT intend that – now removed.

p. 9 The sentence starts with “Most convincingly” should be unpacked for a clear understanding.

We have rewritten the sentence, thanks.

In Methods, please describe what the participants were told to do.

We have modified the description of what participants were asked to do. For the pupillometry task: “Importantly, participants were instructed to keep their gaze on the fixation point and simply observe the stimuli, without performing any task.” (p 14).

For the psychophysics: “The participant was asked to report whether the first or second array had more dots, by pressing the corresponding key.” (p15).

Please use $\log_{10}BF$ instead of $\lg BF$.

Thank you, we have included all the suggested changes.

REVIEWER COMMENTS

Reviewer #1 (Remarks to the Author):

The authors did a good job addressing my concerns. I have no additional comments.

Reviewer #2 (Remarks to the Author):

The authors have done an excellent job with the revision and responded satisfactorily to all my queries, also by adding an experiment.

Reviewer #3 (Remarks to the Author):

I appreciate the additional experiment conducted to address my concern. All of my concerns were adequately addressed and I have no further concerns.